# Circadian patterns of heart rate variability in fetal sheep after hypoxia-ischaemia: A biomarker of evolving brain injury

Christopher A. Lear[1] 🆔 , Yoshiki Maeda[1,2], Victoria J. King[1], Simerdeep K. Dhillon[1], Michael J. Beacom[1], Mark I. Gunning[1], Benjamin A. Lear[1] 🆔 , Joanne O. Davidson[1] 🆔 , Peter R. Stone[3] 🆔 , Tomoaki Ikeda[2], Alistair J. Gunn[1] 🆔 and Laura Bennet[1] 🆔

[1]*Department of Physiology, Fetal Physiology and Neuroscience Group, The University of Auckland, Auckland, New Zealand*
[2]*The Department of Obstetrics and Gynaecology, Mie University, Mie, Japan*
[3]*The Department of Obstetrics and Gynaecology, The University of Auckland, Auckland, New Zealand*

Handling Editors: Peter Kohl & Michael Shattock

The peer review history is available in the Supporting Information section of this article (https://doi.org/10.1113/JP284560#support-information-section).

**Abstract** Hypoxia-ischaemia (HI) before birth is a key risk factor for stillbirth and severe neuro-developmental disability in survivors, including cerebral palsy, although there are no reliable biomarkers to detect at risk fetuses that may have suffered a transient period of severe HI. We investigated time and frequency domain measures of fetal heart rate variability (FHRV) for 3 weeks after HI in preterm fetal sheep at 0.7 gestation (equivalent to preterm humans) until 0.8 gestation (equivalent to term humans). We have previously shown that this is associated with delayed

C. A. Lear and Y. Maeda contributed equally to this work.

development of severe white and grey matter injury, including cystic white matter injury (WMI) resembling that observed in human preterm infants. HI was associated with suppression of time and frequency domain measures of FHRV and reduced their circadian rhythmicity during the first 3 days of recovery. By contrast, circadian rhythms of multiple measures of FHRV were exaggerated over the final 2 weeks of recovery, mediated by a greater reduction in FHRV during the morning nadir, but no change in the evening peak. These data suggest that the time of day at which FHRV measurements are taken affects their diagnostic utility. We further propose that circadian changes in FHRV may be a low-cost, easily applied biomarker of antenatal HI and evolving brain injury.

(Received 21 February 2023; accepted after revision 23 June 2023; first published online 30 June 2023)
**Corresponding author** L. Bennet: Department of Physiology, Faculty of Medical and Health Sciences, The University of Auckland, Private Bag 92019 Auckland 1142, New Zealand.     Email: l.bennet@auckland.ac.nz

**Abstract figure legend** Hypoxia-ischaemia (HI) at preterm gestation triggers evolving white and grey matter injury and is associated with stillbirth and lifelong disability in survivors. We studied the longitudinal patterns of fetal heart rate variability (FHRV) during recovery from HI until term equivalent gestation in fetal sheep to assess whether it would provide a biomarker of evolving brain injury. Changes in frequency domain measures of FHRV (VLF, very low frequency; LF, low frequency; HF, high frequency) are shown. Circadian FHRV rhythms were suppressed for up to 1 week after HI. However, exaggerated circadian FHRV rhythms were observed between 2 and 3 weeks after HI, with notably deeper morning nadirs of FHRV. These alterations in the circadian rhythmicity of FHRV may represent biomarkers of evolving brain injury after HI. Created with Biorender.com.

## Key points

- Hypoxia-ischaemia (HI) before birth is a key risk factor for stillbirth and probably for disability in survivors, although there are no reliable biomarkers for antenatal brain injury.
- In preterm fetal sheep, acute HI that is known to lead to delayed development of severe white and grey matter injury over 3 weeks, was associated with early suppression of multiple time and frequency domain measures of fetal heart rate variability (FHRV) and loss of their circadian rhythms during the first 3 days after HI.
- Over the final 2 weeks of recovery after HI, exaggerated circadian rhythms of frequency domain FHRV measures were observed. The morning nadirs were lower with no change in the evening peak of FHRV.
- Circadian changes in FHRV may be a low-cost, easily applied biomarker of antenatal HI and evolving brain injury.

## Introduction

Hypoxia-ischaemia (HI) is a key contributor to perinatal brain injury and lifelong neurodevelopmental disability (Back, 2017; Dhillon et al., 2018; Manuck et al., 2016). The World Health Organization highlighted that, in 2020,

∼5 million children under the age of 5 years died, mostly from preventable and treatable causes, of whom around half were newborns (World Health Organization, 2022). In addition to these cases, there is evidence that a significant proportion of HI at both term and preterm gestation occurs before birth (Nakao et al., 2022,

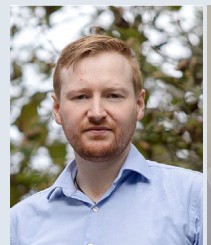
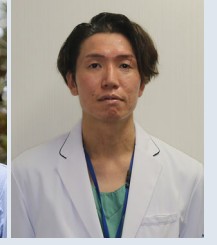

**Christopher Lear** is a developing clinician-scientist working as a Senior Research Fellow in the Fetal Physiology and Neuroscience Group, University of Auckland and a Trainee Intern at Auckland City Hospital, New Zealand. His goal is to translate enhanced understanding of fundamental fetal physiology and improve the clinical identification of fetuses at risk of hypoxic-ischaemic brain injury. **Yoshiki Maeda** is an obstetrician at Kuwana City Medical Centre, Japan. His research, in collaboration with Professors Laura Bennet and Alistair Gunn at the University of Auckland, seeks to improve fetal heart rate monitoring, aiming to improve the diagnosis of prenatal hypoxia. He considers that this is key to developing new treatment strategies and improving obstetric outcomes.

2023). For example, 7.3% of the preterm infants who developed cerebral palsy and 26.9% of those who died within 24 months had cerebral appearances on imaging consistent with an antenatal ischaemic or haemorrhagic injury (de Vries et al., 1998). Moreover, EEG patterns in preterm infants with cystic white matter injury (WMI) suggested antenatal injury in ∼19% of cases (Hayakawa et al., 1999). Critically, 88% of preterm and term stillbirths after 20 weeks of gestation were attributed to HI (Pacora et al., 2019), whereas 46% of early neonatal deaths ≥24 weeks of gestation showed brain injury that was consistent with antenatal onset (Becher et al., 2004; Bell et al., 2005). Thus, many cases of stillbirth and early neonatal death probably represent 'silent' cases of antenatal HI.

Severe, cystic WMI occurs in 1−3% of surviving preterm infants (Ancel et al., 2015) and is associated with an ∼15–30-fold increased risk of cerebral palsy (Rees et al., 2022), suggesting that it is the underlying pathological substrate for severe disability (Lear et al., 2022). We have recently reported that severe HI at 0.7 gestation in preterm fetal sheep is associated with slow evolving cystic WMI, which only became evident between 14 and 21 days after HI (Lear et al., 2021; Lear, Lear, Davidson et al., 2023) in addition to grey matter injury (Lear, Lear, Dhillon et al., 2023). The histological characteristics and delayed evolution of injury were highly similar to cystic WMI described in preterm infants (Back, 2017; Pierrat et al., 2001). These experimental observations support a direct link between antenatal HI at preterm gestation, cystic WMI and cerebral palsy in survivors (Lear et al., 2022).

To reduce the global burden of perinatal mortality and morbidity (Perin et al., 2022; World Health Organization, 2022), we need new diagnostic and prognostic biomarkers that can be used both prenatally and postnatally, and can be readily adapted for use in low-resource settings. There are still limited ways to identify fetuses who have been exposed to and survived antenatal HI. Fetal surveillance is largely limited to infrequent assessment of fetal growth via ultrasound or fetal heart rate (FHR) patterns and derivative indices such as FHR variability (FHRV). However, the effect of antenatal HI on FHR patterns remains poorly studied (Lear et al., 2018).

Recent studies have evaluated FHR and FHRV patterns in conjunction with magnetic resonance imaging analysis of brain injury in a Japanese cohort of cerebral palsy cases (Nakao et al., 2020, 2022, 2023). These studies demonstrated that, although intrapartum HI contributes to brain injury and cerebral palsy, a significant number of babies (57% of preterm and 30% of near-term and term fetuses) experience HI before labour, leading to established neural injury, including basal ganglia-thalamic and WMI. Injury patterns and the timing of insults are associated with different patterns of abnormal FHR, including persistent bradycardia and non-reassuring

patterns (Nakao et al., 2020, 2022, 2023). Previous work by Fukushima et al. (2016) also identified antenatal causes for many cases of cerebral palsy, and showed reduced FHRV, with loss of fetal body movements being common within 1 week of HI, but less common thereafter as injury progressed. These limited data suggest that FHR and FHRV patterns have potential utility as a biomarker for antenatal brain injury.

In the present study, we investigated the long-term changes in FHR patterns in this paradigm of slowly evolving cystic WMI after severe HI (Lear et al., 2021), aiming to test the hypothesis that antenatal HI would be associated with altered FHRV that may serve as a biomarker. Preterm fetal sheep received HI at 0.7 gestation and were studied for 3 weeks post-HI. Sheep are highly precocial and so this interval broadly spans the equivalent brain maturation of humans between 30 weeks and full term (Barlow, 1969; de Graaf-Peters & Hadders-Algra, 2006; McIntosh et al., 1979). Here, we provide evidence that antenatal HI at preterm equivalent is indeed associated with abnormal FHRV, which was strikingly observed in a circadian-dependent manner.

## Methods

### Ethical approval

All procedures were approved by the Animal Ethics Committee of the University of Auckland (number 22 069) following the New Zealand Animal Welfare Act 1999, as well as the Code of Ethical Conduct for animals in research established by the Ministry of Primary Industries, Government of New Zealand, and also comply with the ARRIVE guidelines (Percie du Sert et al., 2020).

### Surgical procedures

Eighteen singleton Romney/Suffolk fetal sheep were surgically instrumented at 98−100 days of gestation (term = 147 days), as previously reported (Lear et al., 2021). Food but not water was withdrawn 12−18 h before surgery. Ewes were given long acting oxytetracycline (20 mg kg$^{-1}$; Phoenix Pharm Distributors, Auckland, New Zealand) I.M. 30 min before surgery for prophylaxis. Anaesthesia was induced by I.V. injection of propofol (5 mg kg$^{-1}$; AstraZeneca, Auckland, New Zealand) and general anaesthesia was maintained using 2−3% isoflurane in oxygen. The depth of anaesthesia was constantly monitored by trained anaesthetic staff. Ewes received a slow infusion of isotonic saline (∼250 mL h$^{-1}$) to maintain fluid balance.

A midline abdominal incision was made to expose the uterus, and the fetus was partially exteriorised for instrumentation. Polyvinyl catheters (SteriHealth; Dandenong South, VIC, Australia) were placed in the

left saphenous artery to measure blood pressure and right brachial artery for pre-ductal blood sampling. An additional catheter was placed into the amniotic sac for the measurement of amniotic fluid pressure. A pair of electrodes (AS633-3SSF; Cooner Wire, Chatsworth, CA, USA) was placed s.c. over the right shoulder and at the level of the fifth intercostal space to measure the fetal ECG from which FHR and FHRV were derived. Two pairs of electrodes (AS633-7SSF; Cooner Wire) were placed over the parietal dura bilaterally, 5 and 10 mm anterior to bregma and 10 mm lateral to the sagittal suture for the measurement of EEG activity. A further two electrodes (AS633-7SSF; Cooner Wire) were sewn into nuchal muscle for measurement of nuchal EMG activity. A reference electrode was sewn over the occiput. An inflatable silicone occluder (OC18HD; In Vivo Metric, Healdsburg, CA, USA) was placed around the umbilical cord.

The fetus was returned to the uterus, gentamicin (80 mg; Pfizer, Auckland, New Zealand) was administered to the amniotic sac and the uterus was closed. The maternal midline skin incision was infiltrated with long-acting analgesic (10 mL of 0.5% bupivacaine plus adrenaline; AstraZeneca). All fetal leads were exteriorised through the maternal flank, the maternal peritoneum and skin were closed, and a maternal long saphenous vein was catheterised for postoperative care.

### Postoperative care

After recovery from anaesthesia, ewes were housed in separate metabolic cages with companion animals under a 12 h light/dark photocycle (lights on 06.00 h) at $16 \pm 1°C$ and $50 \pm 10\%$ relative humidity with *ad libitum* access to water and food. Ewes received i.v. antibiotics daily for 4 days (600 mg of benzylpenicillin sodium; Novartis, Auckland, New Zealand; 80 mg of gentamicin, Pfizer). Fetal vascular catheters were maintained patent by continuous infusion of heparinised saline (20 U mL$^{-1}$ at 0.2 mL h$^{-1}$). Twenty-one days after the start of experiment, the ewes and their fetuses were killed by an i.v. overdose to the ewe of sodium pentobarbitone (9 g of Pentobarb 300; Chemstock International, Christchurch, New Zealand); this method is fully consistent with the Animal Welfare Act of New Zealand. Fetal sex was confirmed and organ weights were measured.

### Experimental recordings

Fetal arterial pressure, ECG, EEG activity and nuchal EMG were recorded continuously from 24 h before until 21 days after HI for offline analysis using custom LabVIEW-based data acquisition programs (National Instruments, Austin, TX, USA). Arterial pressure was recorded using Novatrans III Gold pressure transducers

(MX860; Medex, Hilliard, OH, USA) and corrected for maternal movement by subtraction of amniotic fluid pressure. The pressure signals were amplified $500\times$, low-pass filtered with a Butterworth filter set at 20 Hz and saved at 64 Hz, from which mean arterial pressure (MAP) was calculated. The raw ECG signal was analogue filtered with a first-order high-pass filter set at 1 Hz and an eighth-order low-pass Bessel filter set at 100 Hz and saved at 1024 Hz, and used to derive FHR and FHRV (Lear et al., 2016). The EEG signal was low-pass filtered through an inverse Chebyshev filter set at 128 Hz and saved at 256 Hz. EEG power was derived from the power spectrum signal between 0.5 and 20 Hz. The nuchal EMG signal was band-pass filtered between 100 Hz and 1 kHz, then integrated using a time constant of 0.1 s and digitalised at 512 Hz.

### Experimental protocol

Experiments began at 09.00 h, 4−5 days after surgery when fetuses were at 104−105 days of gestation. Fetuses were randomly assigned to either sham occlusion (control, $n = 9$) or HI ($n = 9$) (Lear et al., 2021). Reliable ECGs could not be recorded from two fetuses from each group, which were thus excluded from this study, leaving final numbers of: control ($n = 7$) and HI ($n = 7$). For comparison with our previous dataset, the two fetuses excluded from the HI group were from the subgroup that developed marked white matter atrophy and ventriculomegaly as their predominant macroscopic pattern of WMI (Lear et al., 2021). The ECG of one control fetus failed on day 13, and its ECG-based outcomes were excluded from this point onwards (leaving control $n = 6$ for the final week of recordings). MAP recordings failed in one HI fetus on day 2, which was excluded from this point onwards, leaving a final number of HI ($n = 6$) for the remainder of the experiment.

HI was induced via complete UCO for 25 min by inflating the umbilical cord occluder with a volume of saline known to completely occlude the umbilical cord (Bennet et al., 1999). The occluder was completely deflated after 25 min. Successful occlusion was confirmed by the rapid onset of bradycardia, a rise in MAP and changes in pH and blood gas measurements (Bennet et al., 1999). The occluder was not inflated in the sham control group.

### Data analysis

Offline analysis of non-ECG derived physiological data was performed using custom LabVIEW-based programs (National Instruments) and shown as hourly means. For ECG derived measures, continuous RR intervals were extracted from the fetal ECG using LabVIEW software and imported into Kubios Standard HRV, version 3.0.2 (Kubios, Kuopio, Finland). All indices were assessed over

5-min contiguous epochs and subsequently averaged to hourly means. Time domain measures of the standard deviation of RR intervals (SDNN) and the root mean squared of successive RR interval differences (RMSSD) were calculated. SDNN was calculated as the SD of all RR intervals during each epoch, providing a measure of total FHRV irrespective of the frequency of oscillations (Task Force of the European Society of Cardiology & the North American Society of Pacing & Electrophysiology, 1996). RMSSD was calculated as the root mean square of successive RR intervals during each epoch, providing a measure of beat-to-beat FHRV, which is sensitive to high frequency oscillations (Lear et al., 2016). Frequency domain analysis was performed by fast Fourier transform spectrum using Welch's periodogram using an interpolation rate of 4 Hz and 50% window overlap. The frequency bands interrogated included the standard adult bands: very low frequency (VLF), 0–0.04 Hz; low frequency (LF); 0.04–0.15 Hz; high frequency (HF), 0.15–0.4 Hz (Task Force of the European Society of Cardiology & the North American Society of Pacing & Electrophysiology, 1996), as well as the extended fetal HF band (extended-HF, 0.15–1.0 Hz) (Siira et al., 2005). VLF, LF, HF and extended-HF are shown as powers.

The number, amplitude and duration of accelerations of the FHR trace were analysed by visual inspection of the FHR trace (shown as continuous 1-s means), by a single obstetrically trained investigator (YM) who was blinded to group by individual coding of data files. This analysis was performed on fetuses with reliable ECG recordings over the entire experimental period: control ($n = 6$) and HI ($n = 7$). Accelerations were defined in accordance with the FIGO guidelines for human fetuses <32 weeks of gestational as abrupt (onset of acceleration to peak FHR in <30 s) increases in FHR above the baseline, of more than 10 bpm and lasting >10 s but less than 10 min (Ayres-de-Campos et al., 2015). Accelerations were assessed during 2-h epochs in both the morning (08.00 h to 10.00 h) and evening (20.00 h to 22.00 h) at three separate time points: 1, 2 and 3 weeks of recovery after HI.

Circadian rhythms were assessed separately over week 2 (days 8−14) and week 3 (days 15−21) of recovery after HI. All parameters assessed showed circadian rhythms with morning nadirs and evening peaks. For each parameter, the daily magnitude and timing of the nadir and peak were identified from hourly means, and the peak-to-peak amplitude was calculated as the peak magnitude minus nadir magnitude. Daily values were then converted into weekly averages.

## Statistical analysis

Statistical analysis was performed using SPSS, version 25 (IBM Corp., Armonk, NY, USA). The effect of HI was

evaluated using two-way analysis of variance (ANOVA) with time treated as a repeated measure and group as the independent factor. Time of day was included as an independent factor when assessing the effect of HI on accelerations. Individual time points were then assessed using one-way ANOVA. Data are presented as the mean ± SD. $P < 0.05$ was considered statistically significant.

## Results

### Baseline parameters

Before the start of experiments, all fetuses were healthy based on our laboratory standards including normal physiological and arterial blood gas parameters. There were no significant differences in any physiological parameters between the groups (Table 1). All fetuses were singletons. The distribution of female and male fetuses in each group was similar: control (3 female, 4 male) and HI (5 female, 2 male).

### Umbilical cord occlusion

Umbilical cord occlusion (UCO) was used to induce HI and was associated with sustained bradycardia and severe hypotension. The nadir of MAP during UCO was $10.2 \pm 1.3$ mmHg in the HI group and $35.4 \pm 0.3$ mmHg during sham-UCO in the control group. UCO was associated with significant hypoxemia, hypoglycaemia and progressive respiratory and metabolic acidosis (Tables 1 and 2).

### Physiological recovery after HI

A significant maturational increase in EEG power ($P < 0.0001$) and MAP ($P = 0.0004$) (Fig. 1) over the 3-week recovery period was observed in both groups, with a progressive maturational fall in FHR ($P < 0.0001$) (Fig. 2). HI was associated with a significant reduction in FHR from 13 to 72 h after HI ($P = 0.0434$) and a significant increase in MAP from 0 to 48 h of recovery (0–6 h, $P = 0.0173$; 7−12 h, $P = 0.0019$; 13−48 h, $P = 0.0413$). After these time points, there were no significant differences between groups in FHR or MAP. HI was associated with reduced EEG power over the entire recovery period (0–6 h, $P < 0.0001$; 7−12 h, $P < 0.0001$; 13−72 h, $P < 0.0001$; 73−168 h, $P = 0.0023$; 169−336 h, $P = 0.0138$; 337−504 h, $P = 0.0284$).

Nuchal EMG activity, an index of fetal body movements, was decreased from 7 to 12 h after HI ($P = 0.0004$) and subsequently increased from 73 h until the end of recovery (73–168 h, $P = 0.0012$; 169−336 h, $P = 0.0024$; 337−504 h, $P = 0.0135$) (Fig. 1). Nuchal

**Table 1. Fetal biochemistry during HI and early recovery.**

| | Group | Baseline | P | UCO (5 min) | P | UCO (17 min) | P | +2 h | +4 h | +6 h | P |
|---|---|---|---|---|---|---|---|---|---|---|---|
| pH | Control | 7.36 ± 0.02 | 0.9368 | 7.36 ± 0.02 | <0.0001 | 7.36 ± 0.03 | <0.0001 | 7.36 ± 0.02 | 7.35 ± 0.02 | 7.36 ± 0.01 | 0.8735 |
| | HI | 7.35 ± 0.03 | | **7.04 ± 0.03** | | **6.84 ± 0.02** | | 7.31 ± 0.05 | 7.39 ± 0.03 | 7.40 ± 0.02 | |
| $P_{aCO_2}$ (mmHg) | Control | 48.0 ± 2.0 | 0.1207 | 45.9 ± 2.1 | <0.0001 | 46.0 ± 2.0 | <0.0001 | 47.1 ± 3.0 | 44.9 ± 3.4 | 48.4 ± 2.3 | 0.0684 |
| | HI | 50.6 ± 4.0 | | **99.2 ± 6.4** | | **138.9 ± 9.0** | | 47.9 ± 1.9 | 47.1 ± 3.3 | 48.5 ± 2.3 | |
| $P_{aO_2}$ (mmHg) | Control | 25.5 ± 1.8 | 0.2224 | 24.7 ± 2.1 | <0.0001 | 24.2 ± 2.0 | <0.0001 | 25.6 ± 2.4 | 25.1 ± 1.4 | 25.8 ± 2.3 | 0.9159 |
| | HI | 24.5 ± 2.5 | | **6.5 ± 0.8** | | **8.9 ± 2.8** | | 26.9 ± 4.0 | 24.0 ± 2.7 | 25.7 ± 3.3 | |
| Hct (%) | Control | 26.8 ± 2.5 | 0.6895 | 26.0 ± 2.8 | | 26.0 ± 3.0 | 0.1479 | 26.3 ± 2.5 | 24.7 ± 2.5 | 25.6 ± 3.0 | 0.0955 |
| | HI | 26.5 ± 2.4 | | 28.2 ± 3.2 | | 28.9 ± 1.3 | | 28.3 ± 2.1 | 27.9 ± 1.3 | 28.2 ± 2.2 | |
| $O_2$ct (mmol L$^{-1}$) | Control | 3.6 ± 0.3 | 1.0000 | 3.4 ± 0.4 | <0.0001 | 3.3 ± 0.5 | <0.0001 | 3.5 ± 0.3 | 3.2 ± 0.5 | 3.4 ± 0.5 | 0.0511 |
| | HI | 3.6 ± 0.3 | | **0.4 ± 0.2** | | **0.5 ± 0.2** | | 4.4 ± 1.0 | 3.8 ± 0.5 | 4.2 ± 0.5 | |
| Lactate (mmol L$^{-1}$) | Control | 0.9 ± 0.2 | 0.5013 | 0.9 ± 0.1 | <0.0001 | 0.9 ± 0.2 | <0.0001 | 0.9 ± 0.1 | 0.9 ± 0.1 | 1.0 ± 0.1 | **0.0002** |
| | HI | 0.8 ± 0.1 | | **3.7 ± 0.4** | | **6.0 ± 0.5** | | **4.1 ± 2.0** | **2.5 ± 1.4** | **1.8 ± 0.6** | |
| Glucose (mmol L$^{-1}$) | Control | 1.1 ± 0.2 | 0.4732 | 1.0 ± 0.1 | <0.0001 | 1.0 ± 0.1 | <0.0001 | 1.2 ± 0.2 | 1.1 ± 0.2 | 1.2 ± 0.2 | **0.0152** |
| | HI | 0.9 ± 0.2 | | **0.4 ± 0.3** | | **0.6 ± 0.3** | | 1.3 ± 0.2 | 1.3 ± 0.2 | 1.4 ± 0.2 | |

Fetal arterial pH, blood gases and metabolites during and after sham occlusion (control, $n = 7$) or HI ($n = 7$) induced by umbilical cord occlusion. $P_{aCO_2}$, arterial pressure of carbon dioxide; $P_{aO_2}$, arterial pressure of oxygen; Hct, haematocrit; $O_2$ct; arterial oxygen content. Data are the mean ± SD. Statistical analysis comprised ANOVA with group as the independent factor and time as repeated measure; baseline was assessed by one-way ANOVA. Significant effects of group are shown in bold.

EMG activity in the control group was characterised by episodic activity interspersed with periods of atonia throughout the 3-week recovery period. By contrast, the increased nuchal EMG activity after HI was predominantly mediated by increased background EMG activity, with reduced frequency of atonia.

## Recovery of fetal heart rate variability after HI

There was a maturational increase over the 3-week recovery period in both groups in SDNN ($P < 0.0001$), RMSSD ($P = 0.0001$) (Fig. 2), VLF power ($P < 0.0001$), LF power ($P = 0.0004$), HF power ($P = 0.0001$) and extended-HF power ($P < 0.0001$) (Fig. 3). No

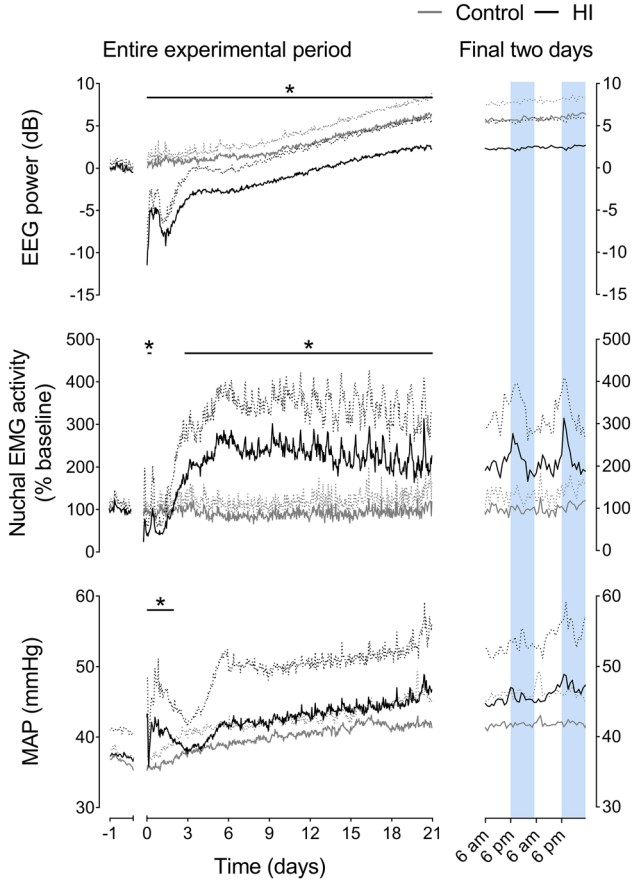

**Figure 1. EEG power, nuchal EMG activity and MAP after HI**
Left: time sequence of changes in EEG power, nuchal EMG activity and MAP from 1 day before until 21 days after sham occlusion (control, grey, $n = 7$) or HI (black, $n = 7$ for EEG power and nuchal EMG; from day 2−21 $n = 6$ for MAP) induced by umbilical cord occlusion. The period of occlusion is not shown, data are shown as the 1-h mean ± SD (SD shown as dotted lines). Statistical analysis comprised ANOVA with group as the independent factor and time as repeated measure. *$P < 0.05$, effect of HI. For exact $p$ values, see text and Statistical Summary Document. Right: enlargement of the final 48 h of recovery for each parameter. Shading represents the hours of darkness (18.00 h to 06.00 h). [Colour figure can be viewed at wileyonlinelibrary.com]

**Table 2. Fetal biochemistry during late recovery.**

| | Group | +1 day | +3 days | P | +7 days | +14 days | +21 days | P |
|---|---|---|---|---|---|---|---|---|
| pH | Control | 7.35 ± 0.03 | 7.35 ± 0.03 | 0.0721 | 7.36 ± 0.02 | 7.34 ± 0.02 | 7.33 ± 0.03 | 0.1550 |
| | HI | 7.36 ± 0.04 | 7.37 ± 0.03 | | 7.37 ± 0.04 | 7.35 ± 0.03 | 7.37 ± 0.03 | |
| $P_{aCO_2}$ (mmHg) | Control | 48.4 ± 2.3 | 47.5 ± 1.5 | 0.2528 | 48.3 ± 2.6 | 49.0 ± 2.9 | 49.9 ± 1.6 | 0.3990 |
| | HI | 47.0 ± 4.5 | 46.9 ± 5.1 | | 49.9 ± 5.8 | 51.6 ± 4.3 | 47.6 ± 2.8 | |
| $P_{aO_2}$ (mmHg) | Control | 25.9 ± 2.8 | 26.8 ± 2.4 | **0.0018** | 25.3 ± 3.7 | 25.8 ± 4.4 | 22.3 ± 4.8 | **0.0454** |
| | HI | **29.0 ± 3.2** | **30.8 ± 3.8** | | **28.8 ± 4.6** | **28.0 ± 3.5** | **27.6 ± 3.6** | |
| Hct (%) | Control | 26.7 ± 3.0 | 30.0 ± 5.0 | 0.4371 | 32.2 ± 6.6 | 32.0 ± 4.2 | 32.0 ± 1.4 | 0.6318 |
| | HI | 29.4 ± 1.6 | 30.7 ± 7.4 | | 31.0 ± 3.8 | 31.9 ± 3.7 | 31.0 ± 2.9 | |
| $O_2$ct (mmol L$^{-1}$) | Control | 3.7 ± 0.6 | 3.9 ± 0.3 | **0.0004** | 3.8 ± 0.7 | 4.0 ± 0.8 | 3.3 ± 1.0 | 0.7939 |
| | HI | **4.5 ± 0.4** | **4.6 ± 0.5** | | 4.7 ± 0.6 | 4.5 ± 0.4 | 4.5 ± 0.1 | |
| Lactate (mmol L$^{-1}$) | Control | 0.9 ± 0.2 | 0.9 ± 0.1 | 0.0971 | 0.9 ± 0.1 | 0.8 ± 0.1 | 1.2 ± 0.4 | **0.0109** |
| | HI | 1.4 ± 0.7 | 0.8 ± 0.2 | | **0.7 ± 0.1** | **0.7 ± 0.1** | **0.8 ± 0.1** | |
| Glucose (mmol L$^{-1}$) | Control | 1.1 ± 0.2 | 1.0 ± 0.2 | 0.0837 | 1.0 ± 0.3 | 0.7 ± 0.2 | 0.8 ± 0.2 | 0.1002 |
| | HI | 1.4 ± 0.3 | 1.1 ± 0.2 | | 0.9 ± 0.2 | 0.8 ± 0.2 | 0.9 ± 0.1 | |

Fetal pH, blood gases and metabolites during late recovery after sham occlusion (control, *n* = 7) or HI (*n* = 7) induced by umbilical cord occlusion. $P_{aCO_2}$, arterial pressure of carbon dioxide; $P_{aO_2}$, arterial pressure of oxygen; Hct, haematocrit; O2ct; arterial oxygen content. Data are the mean ± SD. Statistical analysis comprised ANOVA with group as the independent factor and time as repeated measure. Significant effects of group are shown in bold.

maturational change in LF/HF ratio was observed ($P = 0.1650$, data not shown).

HI was associated with a decrease in SDNN ($P = 0.0163$), VLF power ($P = 0.0009$) and LF power ($P = 0.0359$) from 0 to 6 h after HI. No change in RMSSD was observed at this time ($P = 0.4098$), whereas HF power ($P = 0.0092$) and extended-HF power ($P = 0.0407$) were only reduced from 0 to 2 h after HI.

All indices of FHRV returned to control values between 6 and 12 h after HI. From 13 h after HI until the end of experiment, HI was associated with reduced SDNN (13–72 h, $P = 0.0018$; 73−168 h, $P = 0.0215$; 169−336 h, $P = 0.0217$; 337−504 h, $P = 0.0229$), RMSSD (13–72 h, $P = 0.0017$; 73−168 h, $P = 0.0006$; 169−336 h, $P = 0.0004$; 337−504 h, $P = 0.0141$) and VLF power (13–72 h, $P = 0.0020$; 73−168 h, $P = 0.0423$; 169−336 h, $P = 0.0485$; 337−504 h, $P = 0.0262$). Both LF power ($P = 0.0175$) and HF power ($P = 0.0069$) were reduced in the HI group compared to controls from 13 to 72 h after HI. HF was thereafter not significantly different from controls. Extended-HF power was decreased in the HI group from 13 to 335 h (13–72 h, $P = 0.0029$; 73−168 h, $P = 0.0230$; 169−336 h, $P = 0.0338$) but was no different from controls over the final week of recovery ($P = 0.1081$). LF power was later reduced in the HI group compared to controls over the final week of recovery (336–504 h after HI, $P = 0.0455$). No effect of HI, or an interaction effect between HI or time was observed on the LF/HF ratio at any point during recovery (data not shown). Further statistical results are provided in the Statistical Summary Document.

## Circadian rhythms

In addition to the overall reduction in multiple indices of FHRV (SDNN, RMSSD, VLF, LF and extended-HF powers) during the recovery period, their circadian rhythms were markedly altered after HI. This was characterised by exaggeration of the control circadian pattern, with a greater peak-nadir oscillation, and lower mesor. Overall, the indices of FHRV were lower than control levels during the morning nadir but returned to near-control levels during the evening peak (Figs 2 and 3). Analysis of these circadian rhythms (Table 3) over the final 2 weeks of recovery showed that HI was associated with a reduced and delayed morning nadir in VLF power (magnitude, $P = 0.0154$; time, $P = 0.0002$), LF power (magnitude, $P = 0.0229$; time, $P = 0.0043$), HF power (magnitude, $P = 0.0297$; time, $P = 0.0243$), extended-HF power (magnitude, $P = 0.0119$; time, $P = 0.0634$), RMSSD (magnitude, $P = 0.008$; time, $P = 0.0190$) and SDNN (magnitude, $P = 0.0015$; time, $P = 0.0494$).

The magnitude of the evening peaks in extended-HF power ($P = 0.0253$) and RMSSD ($P = 0.0002$) were lower after HI, whereas the evening peaks in SDNN ($P = 0.0715$), VLF ($P = 0.0630$), LF ($P = 0.1509$) and HF powers ($P = 0.1622$) were not different between the HI and control groups. The timing of the evening peaks was not altered by HI in any parameter. The circadian peak-to-peak amplitude (peak minus nadir) of VLF power ($P = 0.0143$), LF power ($P = 0.0138$) and HF power ($P = 0.0356$) were greater in the HI group than the control group. There was no significant change in extended-HF

power after HI ($P = 0.0724$), whereas the circadian peak-to-peak amplitude of RMSSD was reduced after HI ($P = 0.0036$) (Table 2).

EEG power showed a reduced morning nadir ($P = 0.0225$) and evening peak ($P = 0.0143$) after HI (Table 3). By contrast, the morning nadir ($P = 0.0054$) and evening peak ($P = 0.0032$) of nuchal EMG activity were increased in the HI group. The circadian peak-to-peak amplitudes of FHR ($P = 0.0380$), nuchal EMG activity ($P = 0.0298$) and MAP ($P = 0.0021$) were increased in the HI group (Table 3).

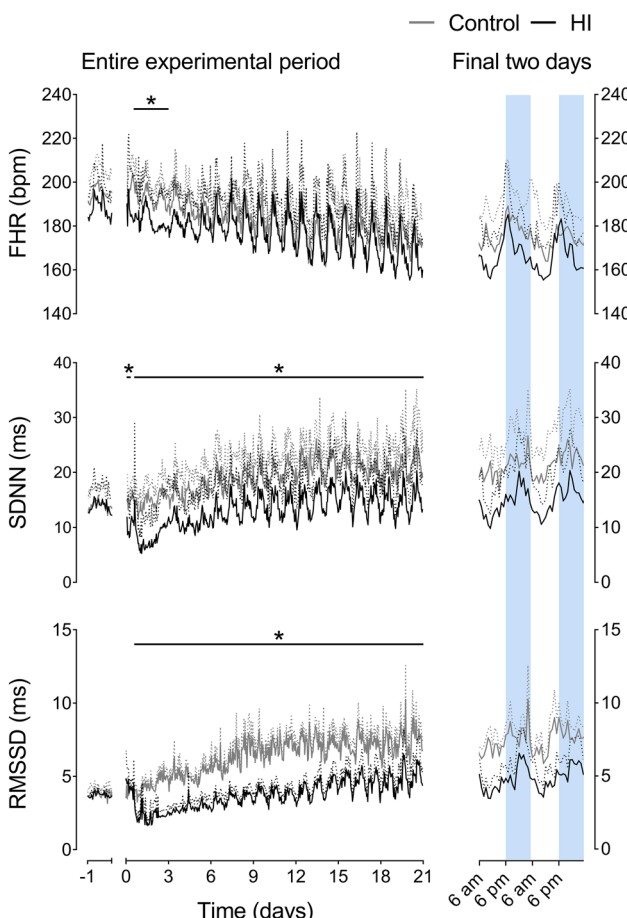

**Figure 2. FHR, SDNN and RMSSD after HI**
Left: time sequence of changes in FHR, SDNN and RMSSD from 1 day before until 21 days after sham occlusion (control, grey, n = 7; day 14–21 n = 6) or HI (black, n = 7) induced by umbilical cord occlusion. The period of occlusion is not shown, data are shown as the 1-h mean ± SD (SD shown as dotted lines). Statistical analysis comprised ANOVA with group as the independent factor and time as repeated measure. *P < 0.05, effect of HI. For exact p values, see text and Statistical Summary Document. Right: enlargement of the final 48 h of recovery for each parameter. Shading represents the hours of darkness (18.00 h to 06.00 h). [Colour figure can be viewed at wileyonlinelibrary.com]

**Table 3. Circadian rhythms during recovery from HI.**

| | Day | Nadir Control | Nadir HI | P | Time of nadir (am) Control | Time of nadir (am) HI | P | Peak Control | Peak HI | P | Time of peak (pm) Control | Time of peak (pm) HI | P | Peak-to-peak amplitude Control | Peak-to-peak amplitude HI | P |
|---|---|---|---|---|---|---|---|---|---|---|---|---|---|---|---|---|
| FHR (bpm) | 7–14 | 171.8 ± 5.7 | 164.3 ± 11.4 | 0.1516 | 8:34 ± 1:47 | 10:04 ± 1:35 | 0.4043 | 201.0 ± 4.8 | 205.4 ± 10.9 | 0.0902 | 7:31 ± 1:17 | 7:31 ± 1:28 | 0.4823 | 29.1 ± 4.9 | 46.7 ± 19.6 | **0.0380** |
| | 14–21 | 163.8 ± 8.3 | 156.2 ± 12.2 | | 9:05 ± 1:10 | 9:58 ± 0:52 | | 195.5 ± 12.7 | 201.5 ± 17.2 | | 8:26 ± 1:17 | 7:19 ± 1:25 | | 31.4 ± 7.6 | 45.3 ± 19.7 | |
| RMSSD (ms) | 7–14 | 4.9 ± 0.6 | **2.6 ± 0.9** | **0.0008** | 6:53 ± 1:08 | **9:35 ± 1:44** | **0.0002** | 9.6 ± 1.2 | **5.0 ± 0.8** | **0.0190** | 7:54 ± 2:09 | 7:44 ± 1:40 | 0.6584 | 4.8 ± 1.4 | **2.5 ± 0.6** | **0.0036** |
| | 14–21 | 5.3 ± 0.9 | **3.0 ± 1.3** | | 9:30 ± 1:11 | **9:31 ± 0:32** | | 9.8 ± 2.1 | **6.4 ± 2.4** | | 7:13 ± 1:16 | 8:08 ± 2:08 | | 4.5 ± 1.6 | **3.4 ± 1.3** | |
| SDNN (ms) | 7–14 | 15.1 ± 2.3 | **9.1 ± 3.5** | **0.0015** | 7:35 ± 1:51 | 9:17 ± 1:11 | 0.0715 | 25.7 ± 2.6 | **20.6 ± 5.4** | **0.0494** | 7:35 ± 0:38 | 7:44 ± 1:32 | 0.8640 | 10.7 ± 2.5 | 11.9 ± 3.7 | 0.2393 |
| | 14–21 | 16.0 ± 2.2 | **9.4 ± 3.0** | | 9:27 ± 1:13 | 9:31 ± 0:44 | | 26.9 ± 4.4 | **22.3 ± 5.2** | | 8:19 ± 1:02 | 8:22 ± 1:59 | | 10.8 ± 3.0 | 12.9 ± 4.2 | |
| Extended-HF (dB) | 7–14 | 2.1 ± 0.3 | **0.9 ± 1.0** | **0.0119** | 7:13 ± 1:14 | **9:37 ± 2:03** | **0.0253** | 3.8 ± 0.3 | 3.0 ± 0.6 | 0.0636 | 8:03 ± 1:09 | 8:03 ± 1:03 | 0.8170 | 1.8 ± 0.3 | 2.1 ± 0.7 | 0.0724 |
| | 14–21 | 2.4 ± 0.2 | **1.1 ± 1.0** | | 9:07 ± 1:41 | **9:37 ± 0:26** | | 3.9 ± 0.4 | 3.2 ± 0.8 | | 8:13 ± 1:12 | 8:00 ± 1:34 | | 1.5 ± 0.4 | 2.1 ± 0.6 | |
| HF (dB) | 7–14 | 1.5 ± 0.3 | **0.3 ± 1.2** | **0.0297** | 7:23 ± 1:16 | 9:53 ± 1:49 | 0.1622 | 3.2 ± 0.3 | **2.6 ± 0.6** | **0.0243** | 8:23 ± 1:59 | 8:19 ± 1:23 | 0.8745 | 1.7 ± 0.2 | **2.3 ± 0.8** | **0.0356** |
| | 14–21 | 1.8 ± 0.3 | **0.6 ± 1.1** | | 9:13 ± 0:56 | 9:35 ± 0:42 | | 3.2 ± 0.5 | **2.9 ± 0.9** | | 7:40 ± 0:56 | 8:06 ± 1:53 | | 1.4 ± 0.3 | **2.3 ± 0.8** | |
| LF (dB) | 7–14 | 3.4 ± 0.2 | **2.3 ± 1.2** | **0.0229** | 7:38 ± 1:32 | 9:44 ± 1:11 | 0.1509 | 4.8 ± 0.4 | **4.4 ± 0.6** | **0.0043** | 7:59 ± 1:34 | 8:36 ± 1:29 | 0.5268 | 1.4 ± 0.3 | **2.1 ± 0.7** | **0.0138** |
| | 14–21 | 3.5 ± 0.2 | **2.5 ± 0.8** | | 8:29 ± 0:37 | 9:35 ± 0:43 | | 4.8 ± 0.4 | **4.4 ± 0.6** | | 7:22 ± 1:39 | 7:47 ± 2:06 | | 1.3 ± 0.3 | **1.9 ± 0.7** | |
| VLF (dB) | 7–14 | 4.4 ± 0.3 | **3.5 ± 0.9** | **0.0154** | 6:54 ± 1:34 | 9:25 ± 0:53 | 0.0630 | 5.8 ± 0.2 | **5.3 ± 0.5** | **0.0002** | 6:55 ± 0:34 | 8:02 ± 1:58 | 0.1867 | 1.4 ± 0.3 | **1.9 ± 0.5** | **0.0143** |
| | 14–21 | 4.6 ± 0.2 | **3.5 ± 0.8** | | 7:41 ± 0:39 | 9:21 ± 0:43 | | 5.8 ± 0.4 | **5.5 ± 0.4** | | 7:24 ± 1:12 | 8:20 ± 1:20 | | 1.2 ± 0.3 | **2.0 ± 0.7** | |
| EEG (dB) | 7–14 | 1.7 ± 1.5 | **−1.7 ± 2.7** | **0.0225** | 4:06 ± 1:11 | **4:48 ± 1:38** | **0.0143** | 2.9 ± 1.2 | −0.6 ± 2.8 | 0.7454 | 5:48 ± 1:14 | 6:54 ± 2:01 | 0.5851 | 1.2 ± 0.6 | 1.1 ± 0.2 | 0.1213 |
| | 14–21 | 4.3 ± 1.9 | **1.1 ± 3.1** | | 6:00 ± 1:41 | **5:48 ± 2:35** | | 5.6 ± 1.8 | 2.0 ± 3.1 | | 7:24 ± 1:37 | 7:06 ± 2:14 | | 1.3 ± 0.2 | 0.9 ± 0.2 | |
| MAP (mmHg) | 7–14 | 38.9 ± 1.6 | 41.3 ± 6.9 | 0.3670 | 7:12 ± 1:54 | 8:32 ± 1:35 | 0.1840 | 41.3 ± 1.6 | 44.4 ± 6.6 | 0.1490 | 7:31 ± 1:33 | 6:57 ± 2:07 | 0.6823 | 2.4 ± 0.5 | **3.2 ± 0.7** | **0.0021** |
| | 14–21 | 40.4 ± 2.4 | 42.9 ± 6.5 | | 6:23 ± 1:47 | 7:33 ± 1:10 | | 43.0 ± 2.5 | 47.4 ± 7.1 | | 7:35 ± 1:24 | 7:30 ± 1:23 | | 2.7 ± 0.5 | **4.5 ± 1.0** | |
| Nuchal (% baseline) | 7–14 | 59.4 ± 11.7 | **201.4 ± 97.3** | **0.0054** | 7:22 ± 1:15 | **6:59 ± 2:23** | **0.0032** | 126.3 ± 25.6 | 302.9 ± 114.9 | 0.5751 | 7:26 ± 0:48 | 6:40 ± 1:13 | 0.5228 | 66.8 ± 22.8 | **101.5 ± 37.9** | **0.0298** |
| | 14–21 | 59.7 ± 13.0 | **157.9 ± 90.3** | | 7:11 ± 0:59 | **6:40 ± 2:22** | | 137.9 ± 43.8 | 299.5 ± 120.1 | | 6:22 ± 1:53 | 7:50 ± 1:07 | | 78.2 ± 35.7 | **141.6 ± 51.1** | |

Magnitude and timing of the daily nadir and peak of each parameter along with the daily peak-to-peak amplitude (i.e. difference between peak and nadir during each 24-h period) after sham occlusion (control) or HI induced by umbilical cord occlusion. For group numbers, see text associated with each parameter. The two timepoints represent the average from 7 days. Data are the mean ± SD; times are shown as hh:mm. Statistical analysis comprised ANOVA with group as the independent factor and time as repeated measure. Significant effects of group are shown in bold.

## Recovery of fetal heart rate accelerations

There was a significant effect of HI and time of day on both the number (HI, $P = 0.0006$; time of day, $P = 0.0021$) and average amplitude (HI, $P = 0.0001$; time of day, $P = 0.0007$) of FHR accelerations over the

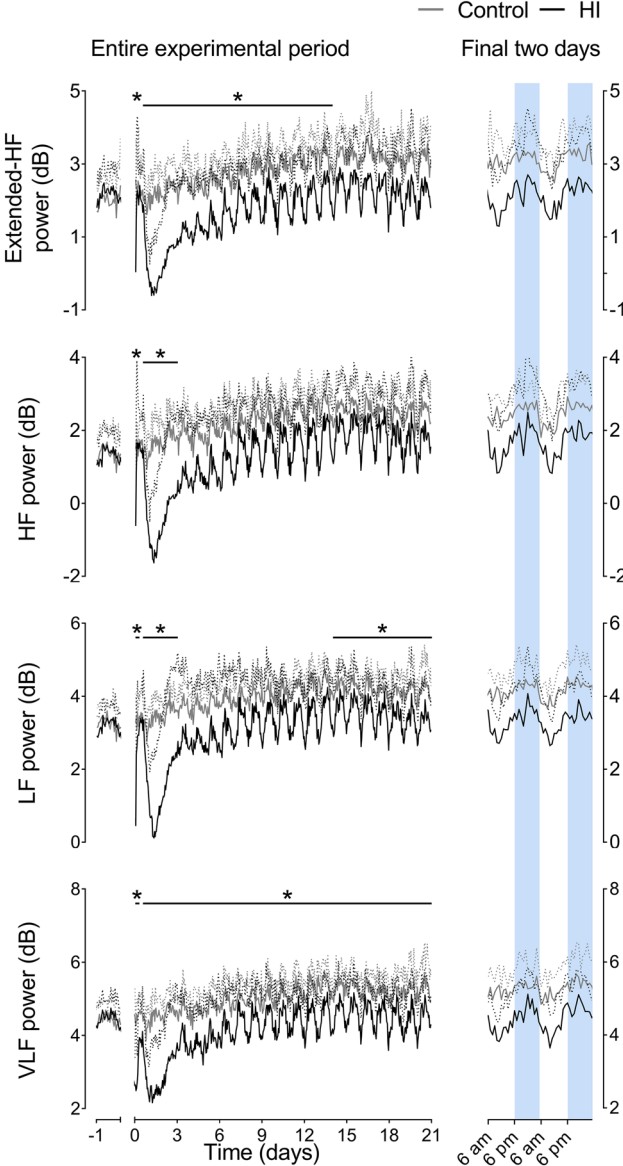

**Figure 3. Extended-HF, HF, LF and VLF powers after HI**
Left: time sequence of changes in extended-HF, HF, LF and VLF from 1 day before until 21 days after sham occlusion (control, grey, n = 7; day 14−21 n = 6) or HI (black, n = 7) induced by umbilical cord occlusion. The period of occlusion is not shown, data are shown as the 1-h mean ± SD (SD shown as dotted lines). Statistical analysis comprised ANOVA with group as the independent factor and time as repeated measure. *P < 0.05, effect of HI. For exact p values, see text and Statistical Summary Document. Right: enlargement of the final 48 h of recovery and each parameter. Shading represents the hours of darkness (18.00 h to 06.00 h). [Colour figure can be viewed at wileyonlinelibrary.com]

3 weeks of recovery (Fig. 4 and Table 4). This suggests that, compared to evening, the morning was associated with fewer and smaller accelerations across both groups and that HI was associated with an additive reduction in both the amplitude and number of accelerations. One-way ANOVA showed that HI was associated with fewer accelerations in the morning after 3 weeks of recovery compared to controls ($P < 0.0001$), but not in the evening ($P = 0.1968$). Neither HI, nor time of day affected the average duration of accelerations.

## Discussion

The present study demonstrates for the first time that FHRV measures provide significant utility as a biomarker for different phases of evolving HI related brain injury *in utero* from preterm to term brain equivalent life. The most striking finding of our results was the significant change to FHRV circadian rhythmicity, with loss or suppression of rhythms for at least 3 days post-HI, followed by progressive development of exaggerated oscillations, with a notably later lower nadir occurring mid-morning. The timing of the most defining feature of the change in oscillations, the nadir, is a key observation because it defines a window of opportunity for routine clinical assessment. An interesting corollary is that an apparently non-reassuring morning FHR trace, including loss of FHR accelerations, followed by apparent improvement over the day and evening, may unexpectedly identify a fetus with evolving or established severe HI injury.

Circadian rhythms are generated by the interplay between both the central clock located in the supra-chiasmatic nucleus (SCN) of the hypothalamus and peripheral clocks (Ruan et al., 2021). The role of circadian rhythmicity is relatively poorly understood before birth, but chronodisruption during pregnancy is associated with adverse health outcomes (Varcoe et al., 2018). The SCN develops early in gestation and fetal circadian rhythmicity is entrained by the placental passage of melatonin (Reiter et al., 2014; Seron-Ferre et al., 2012). It is currently unclear what role the fetal central clock plays (Mark et al., 2017) or how fetal peripheral clocks are regulated (Seron-Ferre et al., 2012). Fetal circadian rhythms have been reported in humans and animals (Bennet et al., 2006; Bennet, Galinsky et al., 2018; Dalton et al., 1977; de Vries et al., 1987; Jensen et al., 2009; Kapaya et al., 2016; Mark et al., 2017; Patrick et al., 1982; Quaedackers et al., 2005; Seron-Ferre et al., 2012; Visser et al., 1982). Diurnal changes in FHR have been observed in human fetuses as young as 20 weeks (de Vries et al., 1987). In fetuses closer to term, FHR and fetal activity (including body and breathing movements) that regulate components of FHRV are known to increase over the day, peaking early to mid-evening, followed by a fall at night and relative quiescence during the morning. These patterns are

**Table 4. Recovery of fetal heart rate accelerations**

| | Weeks of recovery | Morning | | | Evening | | | *P* (HI) | *P* (time of day) |
|---|---|---|---|---|---|---|---|---|---|
| | | 1 | 2 | 3 | 1 | 2 | 3 | | |
| Number (h$^{-1}$) | Control | 17.5 ± 3.5 | 16.8 ± 5.1 | 19.8 ± 4.5 | 18.7 ± 4.6 | 20.0 ± 4.8 | 25.8 ± 7.0 | **0.0006** | **0.0021** |
| | HI | 11.9 ± 11.2 | 10.5 ± 6.7 | 5.5 ± 1.5 | 13.9 ± 6.0 | 18.9 ± 5.3 | 18.9 ± 10.6 | | |
| Average duration (s) | Control | 48.1 ± 5.8 | 49.4 ± 8.0 | 44.9 ± 5.1 | 48.9 ± 4.8 | 50.0 ± 7.3 | 44.6 ± 2.8 | 0.7582 | 0.4033 |
| | HI | 52.8 ± 15.9 | 45.8 ± 9.0 | 36.9 ± 7.6 | 49.9 ± 10.4 | 50.3 ± 6.9 | 45.9 ± 5.7 | | |
| Average amplitude (bpm) | Control | 31.8 ± 3.6 | 33.9 ± 5.2 | 29.3 ± 2.3 | 37.5 ± 5.3 | 44.9 ± 4.6 | 34.5 ± 5.7 | **0.0001** | **0.0007** |
| | HI | 24.4 ± 8.7 | 26.3 ± 5.3 | 21.5 ± 4.9 | 28.3 ± 3.3 | 35.0 ± 8.6 | 27.4 ± 6.6 | | |

Characteristics of FHR accelerations during recovery after sham occlusion (control, *n* = 6) or HI (*n* = 7) induced by umbilical cord occlusion. The number of accelerations is additionally shown in Fig. 4. Data are the mean ± SD. Statistical analysis comprised ANOVA with group and time of day as independent factors and time as repeated measure. Significant effects are shown in bold.

observed similarly in both humans and fetal sheep (Dalton et al., 1977; de Vries et al., 1987; Jensen et al., 2009; Kapaya et al., 2016; Patrick et al., 1982; Quaedackers et al., 2005).

Gestational age is also important, with maturational changes in neural and cardiac development and regulatory control (Lear et al., 2016; Sletten et al., 2018; Tournier et al., 2022). As demonstrated in our control group, there are consistent circadian rhythms for FHR and FHRV around the progressive maturational fall in FHR. This is mediated by maturation of the parasympathetic nervous system alongside increased measures of FHRV, similarly to the human fetus (Sletten et al., 2018; Tournier et al., 2022). FHRV reflects the complex integration of intrinsic pacemaker rhythms of the sinoatrial node with the sympathetic and parasympathetic activity of the autonomic nervous system, modulated by fetal behaviour and sleep state development (Dalton et al., 1983; Frasch et al., 2020; Jensen et al., 2009; Lear et al., 2016; Lear et al., 2020; Tournier et al., 2022). It is important to appreciate

that, during the period investigated in the present study, fetuses transition from expressing all behaviour in a near continuous manner, associated with discontinuous EEG activity, to compartmentalised behaviour associated with the development of sleep-state cycling (Bennet, Walker et al., 2018).

In the present study, we observed two primary changes to circadian rhythmicity after a severe HI insult. The first was the significant suppression or loss of rhythmicity for 72 h after HI, in association with suppression of multiple measures of FHRV (except for transient restoration to control levels at ∼6−12 h after HI). Reduced fetal body movements (Fig. 1) may contribute to loss of a small proportion of FHRV, but not loss of circadian rhythmicity. The loss of circadian rhythmicity is also probably not caused by reduced exposure to maternal melatonin because we have previously shown in this model that fetal plasma levels of melatonin are not changed by HI (Drury et al., 2014), and other zeitgebers such as day/night light

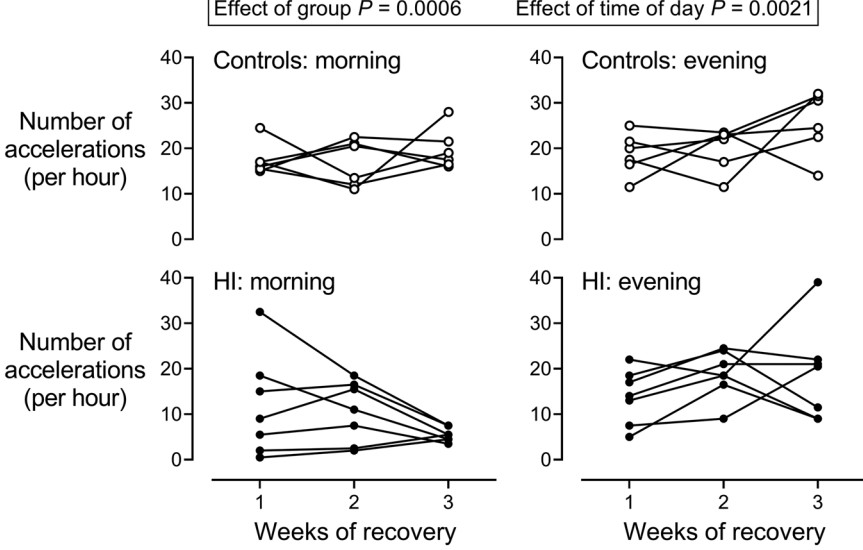

**Figure 4. Fetal heart rate accelerations after HI**
Changes in the number of fetal heart rate accelerations in the morning (08.00h to 10.00 h) and evening (20.00h to 22.00 h) during the 3 weeks of recovery after sham occlusion (control, open circles, *n* = 6) or HI (closed circles, *n* = 7) induced by umbilical cord occlusion. Statistical analysis comprised ANOVA with group and time of day as independent factors and time as repeated measure.

cycles and maternal food intake did not change. However, as discussed below, impairment of mitochondrial function is common after a severe HI insult, and thus, potentially, the SCN may not have been responsive to melatonin.

The early changes in FHR and FHRV align with the known phases of evolving injury after an HI insult (Bennet et al., 2006; Dhillon et al., 2018; Lorek et al., 1994; Yamaguchi et al., 2018). Following initial reperfusion post-HI, there is typically initial restoration of cerebral oxidative metabolism followed by delayed suppression of metabolism in a latent phase of recovery (the first ∼6–5 h of recovery). This phase is characterised by cerebral hypo-perfusion, which is mediated by active neural suppression. This suppression is related to multiple factors including release of endogenous inhibitory neuromodulators and acute neuroinflammation (Bennet, Dhillon et al., 2018; Dhillon et al., 2023). In the present study, after initial suppression, FHR and FHRV largely returned to control values during the latent phase. Following this, there was a phase of secondary deterioration, which is associated with failure of oxidative metabolism despite restoration of oxygenation and bulk cell loss (Bennet et al., 2006; Hagberg et al., 2014; Lorek et al., 1994). The severity of the failure of oxidative metabolism is associated with greater risk of subsequent impaired neurodevelopment (Hope et al., 1984).

The loss of circadian rhythmicity may be related to cell death and consequent loss of cellular control, which would probably impact the functionality of the SCN. Additionally, clock genes regulate mitochondrial function to ensure metabolic demand is aligned to nutrient availability (Manella & Asher, 2016; Peek, 2020). There is some evidence of a reciprocal relationship between mitochondrial function and the clock genes (de Goede et al., 2018; Scrima et al., 2016) and thus mitochondrial dysfunction may impair clock gene rhythmicity. Induction of hypoxia-inducible factor (HIF) proteins may also contribute after HI. HIF plays a key role in metabolism and oxygen homeostasis and, when oxidative capacity is impaired, HIF acts to shift energy use away from oxygen-dependent pathways (Adamovich et al., 2022; O'Connell et al., 2020; Wu et al., 2017). Importantly, HIF facilitates cross-talk between hypoxia and the circadian clock at the level of the genome (Adamovich et al., 2022; O'Connell et al., 2020; Wu et al., 2017). For example, HIF-1$\alpha$ and the circadian clock protein BMAL1 can bind to each other, leading to altered circadian rhythmicity, including the induction of phase changes and the suppression of the amplitude of circadian cycles (Wu et al., 2017). A similar relationship has been reported for the circadian genes *Clock* and *Period* with HIF (Kobayashi et al., 2017; Peek et al., 2017).

Potentially, there may also be a direct effect of hypoxia on the cardiac peripheral clock. In adults, the cardiac clock is in part mediated by circadian changes in the expression of multiple ion channels (Monfredi & Lakatta, 2019), including the Kv1.5 and Kv4.2 potassium channels, which have been shown to demonstrate circadian cyclicity (Yamashita et al., 2003). Cardiac circadian clock control can be altered by ischaemia, and suppressed heart rate variability is a risk factor for cardiac failure (Black et al., 2019). We have previously shown that, after severe global hypoxia in the preterm fetal sheep, there is no evidence of cardiomyocyte injury after 3 days of recovery (Lumbers et al., 2001). However, whether transient cardiac injury occurs that may alter circadian clock control during recovery is unknown.

The second, and perhaps most important observation from the present study was the finding that, when circadian rhythmicity returned after HI, the oscillations were markedly exaggerated for all measures compared to controls as brain injury progressed. Changes were in general characterised by a lower mesor, augmented peak-to-peak amplitude and a notable lower nadir in the morning. The time domain measure RMSSD was particularly affected, with a prolonged reduction compared to control, although circadian cycling was clearer compared to controls. By contrast, FHR itself rapidly returned to control group values. Speculatively, this suggests that circadian rhythmicity is strongly influenced by cardiac peripheral clock activity, whereas FHRV indices may be influenced more by neural control and thus changes in FHRV and not FHR *per se* reflect evolving brain injury.

The return to fetal circadian cycling was progressive after 3 days, and circadian cycling was re-established by 4 days post-HI. This is consistent with resolution of secondary phase events such as seizures (Bennet, Galinsky et al., 2018), as well as progressive recovery of neural function and new cell proliferation (Barrett et al., 2012; Bennet et al., 2006). Potentially, there may also be a contribution from resynchronisation of the circadian clock (Krishnan & Lyons, 2015). In the present study, SDNN, RMSSD and VLF were markedly reduced throughout recovery after HI, from 72 h to 3 weeks. LF returned to control levels from 3 to 14 days, but subsequently fell from 14 to 21 days of recovery, and HF was only suppressed during the first 3 days. These differential findings probably reflect changes in fetal autonomic function and potentially breathing movements. For example, the sustained reduction in RMSSD may reflect altered fetal breathing movements, which are initially supressed after HI, followed by 2−3 days of rapid sustained activity (Yan et al., 2009). This evolution may underpin the variability of RMSDD during the secondary phase. Longitudinal assessment of breathing movements over longterm recovery has not been undertaken. However, in an overlapping cohort of fetuses, we have reported a significant reduction in lung weights at 3 weeks recovery after HI (88.9 $\pm$ 6.4 g in controls

*vs.* 44.3 ± 5.6 g after HI) (Lear et al., 2021). This suggests that breathing movements were suppressed compared to controls, leading to reduced lung growth as observed experimentally and clinically when fetal breathing movements are reduced (Hooper & Harding, 1995),

Fetal body movements were grossly different from controls after the first week, which is consistent with persistent tonic muscle activity with loss of the normal pattern of periods of atonia between discrete movements (Bennet, Walker et al., 2018). This tonic pattern may be related to disrupted corticospinal white matter tracts and elevated levels of serotonin in the spinal cord (Drobyshevsky et al., 2014). Of particular note, there was an exaggerated nocturnal increase in nuchal EMG activity in the HI group, which broadly corresponded with the peak in FHRV. This may have contributed to the peak in FHRV measures being strikingly similar to controls. Changes in body movements, however, did not correlate with the morning nadir.

It is unclear why the circadian rhythms were exaggerated with a more pronounced morning nadir. In addition to changes in autonomic function and fetal behaviour, total cell loss may play a role both within the SCN and other nuclei because the SCN neurons require a neural network of neighbouring cells to function effectively and in a synchronised manner (Azzi et al., 2017). The present experimental paradigm leads to extensive injury to neighbouring subcortical neuronal populations, including thalamus and hippocampus (Lear, Lear, Dhillon et al., 2023), as well as diffuse WMI, with delayed development of white matter cysts (Lear et al., 2021). Injury may cause dysregulation of core clock gene expression, as is seen with acute brain injury where there is perturbation of the key regulators BMAL, Per and Cry (Hetman et al., 2022).

In addition to cell loss, chronic neuroinflammation may alter clock gene expression (Yamakawa et al., 2020). During the secondary phase, acute inflammation facilitates clearance of cellular debris (Hagberg et al., 2015). After resolution of acute inflammation, there may be chronic neuroinflammation that can impair brain maturation and cause ongoing brain injury (Fleiss et al., 2021; Lear et al., 2021; van den Heuij et al., 2019). In adults, inflammation is associated with changes in the circadian clock and rhythmicity and indeed may help entrain the circadian clock (Poole & Ray, 2022). Microglia, the resident macrophages in the brain, also exhibit circadian rhythmicity (Wang et al., 2021). Furthermore, the microglial circadian clock modulates the degree of inflammation (Fonken et al., 2015). We have observed sustained increases in tumour necrosis factor positive cells in the brain and progressive increases in activated microglia in both white and grey matter in our model (Lear et al., 2021; Lear, Lear, Dhillon et al., 2023;

van den Heuij et al., 2019), whereas tumour necrosis factor blockage reduces microglial activation (Lear, Lear, Davidson et al., 2023).

It is important to consider the possibility that altered or impaired circadian rhythms after perinatal HI, as identified in the present study, may contribute to ongoing neurodevelopmental impairment and risk of systemic and metabolic disease. Even in the absence of neural injury, chronodysruption during gestation leads to long-lasting impairment of circadian gene regulation and function in the adrenal gland (Mendez et al., 2019; Richter et al., 2018; Salazar et al., 2018) and hippocampus (Vilches et al., 2014). Gestational chronodysruption is also associated with impaired glucose and adipose tissue regulation, probably increasing the risk of metabolic disease in offspring (Halabi et al., 2021). Moreover, in rodents, gestational chronodysruption has been associated with increased circulating levels of the pro-inflammatory cytokines interleukin-1$\beta$ and interleukin-6 in later life (Mendez et al., 2019). This suggests the hypothesis that persisting chronodysruption may contribute to chronic neuroinflammation and impaired neurodevelopment and, conversely, that re-establishing normal circadian patterns could improve neurodevelopmental and systemic outcomes (Mendez et al., 2022).

Finally, after 3 weeks of recovery from HI, FHR accelerations on the continuous FHR trace were virtually abolished during the morning, whereas accelerations were no different from controls during the evening. Intriguingly, the number of accelerations was observed to progressively fall from 1 to 3 weeks after HI, broadly following the evolution of cystic WMI in the present paradigm (Lear et al., 2021). HI was also associated with a modest reduction in acceleration amplitude, although no change to acceleration duration was observed.

### Significance and perspectives

Reducing perinatal mortality and morbidity is consistent with Goal 3 of the United Nations 2030 Agenda for Sustainable Development (United Nations Department of Economic & Social Affairs, 2022). Antenatal HI is a key contributor to stillbirth and lifelong neurodevelopmental disability among survivors (de Vries et al., 1998; Manuck et al., 2016; Nakao et al., 2022, 2023; Pacora et al., 2019). The present study suggests that suppression followed by an exaggerated circadian rhythm of multiple measures of FHRV, with a delayed and lower morning nadir, may represent useful markers of prior antenatal HI. It may be a particularly useful biomarker of evolving cystic WMI, which is known to be strongly associated with the development of cerebral palsy (Rees et al., 2022). Such biomarkers are urgently needed to better understand the burden of antenatal HI, to identify fetuses at risk of subsequent stillbirth and potentially to identify

fetuses and newborns who might benefit from neuroprotective or neurorestorative interventions. There have been recent advances suggesting neuroprotection of even severe WMI is possible (Lear et al., 2022; Lear, Lear, Davidson et al., 2023). Pragmatically, our data strongly suggest that continuous recordings are probably more useful than single spot recordings, and that the time of day probably impacts the ability to identify fetuses exposed to antenatal HI.

Encouragingly, in the present study, the greatest separation between healthy and injured fetuses was seen on mid-morning recordings. Thus, this 'low-tech', low-cost approach would be easily deployed in both high and low-resource settings. Further work is required to understand the effect of other types and severity of insults. Equally, as discussed, circadian rhythm dysregulation may contribute to ongoing neurodevelopmental impairment and further contribute to an increased risk of systemic disease in later adult life. Understanding the mechanisms mediating the effects of HI and other perinatal complications on circadian rhythms and their downstream effects on health may offer opportunities for neuroprotection and neurorestoration, and to reduce the risk of systemic metabolic disease among survivors of preterm birth.

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

## Additional information

### Data availability statement

Data will be available from the corresponding author on reasonable request.

### Competing interests

The authors declare that they have no competing interests.

### Author contributions

These experiments were conducted in the Fetal Physiology and Neuroscience Group laboratory, at the University of Auckland. C.A.L., A.J.G. and L.B. conceived the hypotheses, experimental design and analysis protocols for this study. C.A.L. and L.B. were responsible for data collection. C.A.L., Y.M., V.K. and M.J.B. performed the physiological analysis. C.A.L. and Y.M. drafted the manuscript and contributed equally to the study and also qualify as equal first authors. All authors were involved in data interpretation, in the editing and revision of the manuscript, approved the final version of the manuscript submitted for publication and agree to be accountable for all aspects of the work. All persons designated as authors qualify for authorship, and all those who qualify for authorship are listed.

### Funding

Manatu Hauora | Health Research Council of New Zealand (HRC): Alistair J. Gunn, Laura Bennet, 22/559; Manatu Hauora | Health Research Council of New Zealand (HRC): Alistair J. Gunn, Laura Bennet, 17/601; Manatu Hauora | Health Research Council of New Zealand (HRC): Laura Bennet, 20/437; Auckland Medical Research Foundation (AMRF): Christopher Arthur Lear, 1 122 002.

### Acknowledgements

Open access publishing facilitated by The University of Auckland, as part of the Wiley - The University of Auckland agreement via the Council of Australian University Librarians.

### Keywords

biomarker, fetal heart rate variability, fetus, hypoxia-ischaemia, preterm

## Supporting information

Additional supporting information can be found online in the Supporting Information section at the end of the HTML view of the article. Supporting information files available:

**Statistical Summary Document**
**Peer Review History**

