## [Peer Review History · The Journal of Physiology]

Circadian patterns of heart rate variability in fetal sheep after hypoxia-ischaemia: a biomarker of evolving brain injury

Christopher Arthur Lear, Yoshiki Maeda, Victoria J. King, Simerdeep Kaur Dhillon, Michael J. Beacom, Mark I. Gunning, Benjamin A. Lear, Joanne Davidson, Peter Richard Stone, Tomoaki Ikeda, Alistair J. Gunn, and Laura Bennet
DOI: 10.1113/JP284560

Corresponding author(s): Alistair Gunn (aj.gunn@auckland.ac.nz)

The following individual(s) involved in review of this submission have agreed to reveal their identity: Claudia Torres-Farfan (Referee #1)

Review Timeline:

Submission Date:	21-Feb-2023
Editorial Decision:	04-Apr-2023
Revision Received:	25-May-2023
Accepted:	23-Jun-2023

Senior Editor: Peter Kohl

Reviewing Editor: Michael Shattock

Transaction Report:

Dear Dr Gunn,

Re: JP-RP-2023-284560 "Altered circadian patterns of heart rate variability in fetal sheep: a biomarker of evolving hypoxic-ischaemic brain injury" by Christopher Arthur Lear, Yoshiki Maeda, Victoria J. King, Simerdeep Kaur Dhillon, Michael J. Beacom, Mark I. Gunning, Benjamin A. Lear, Joanne Davidson, Peter Richard Stone, Tomoaki Ikeda, Alistair J. Gunn, and Laura Bennet

Thank you for submitting your manuscript to The Journal of Physiology. It has been assessed by a Reviewing Editor and by 2 expert referees and we are pleased to tell you that it is potentially acceptable for publication following satisfactory major revision.

LANGUAGE EDITING AND SUPPORT FOR PUBLICATION: If you would like help with English language editing, or other article preparation support, Wiley Editing Services offers expert help, including English Language Editing, as well as translation, manuscript formatting, and figure formatting at www.wileyauthors.com/eoo/preparation. You can also find resources for Preparing Your Article for general guidance about writing and preparing your manuscript at www.wileyauthors.com/eoo/prepresources.

REVISION CHECKLIST:

We look forward to receiving your revised submission.

Yours sincerely,

Peter Kohl
Senior Editor
The Journal of Physiology

REQUIRED ITEMS

- Author photo and profile. First (or joint first) authors are asked to provide a short biography (no more than 100 words for one author or 150 words in total for joint first authors) and a portrait photograph. These should be uploaded and clearly labelled with the revised version of the manuscript. See Information for Authors for further details.
- The Journal of Physiology funds authors of provisionally accepted papers to use the premium BioRender site to create high resolution schematic figures. Follow this link and enter your details and the manuscript number to create and download figures. Upload these as the figure files for your revised submission. If you choose not to take up this offer we require figures to be of similar quality and resolution. If you are opting out of this service to authors, state this in the Comments section on the Detailed Information page of the submission form. The link provided should only be used for the purposes of this submission. Authors will be charged for figures created on this premium BioRender account if they are not related to this manuscript submission.
- A Statistical Summary Document, summarising the statistics presented in the manuscript, is required upon revision. It must be on the Journal's template, which can be downloaded from the link in the Statistical Summary Document section here: https://jp.msubmit.net/cgi-bin/main.plex?form_type=display_requirements#statistics.
- Papers must comply with the Statistics Policy: https://jp.msubmit.net/cgi-bin/main.plex?form_type=display_requirements#statistics.

In summary:

- If $n \leq 30$, all data points must be plotted in the figure in a way that reveals their range and distribution. A bar graph with data points overlaid, a box and whisker plot or a violin plot (preferably with data points included) are acceptable formats.
- If $n > 30$, then the entire raw dataset must be made available either as supporting information, or hosted on a not-for-profit repository e.g. FigShare, with access details provided in the manuscript.
- 'n' clearly defined (e.g. x cells from y slices in z animals) in the Methods. Authors should be mindful of pseudoreplication.
- All relevant 'n' values must be clearly stated in the main text, figures and tables, and the Statistical Summary Document (required upon revision).
- The most appropriate summary statistic (e.g. mean or median and standard deviation) must be used. Standard Error of the Mean (SEM) alone is not permitted.
- Exact p values must be stated. Authors must not use 'greater than' or 'less than'. Exact p values must be stated to three significant figures even when 'no statistical significance' is claimed.
- Statistics Summary Document completed appropriately upon revision.
- A Data Availability Statement is required for all papers reporting original data. This must be in the Additional Information section of the manuscript itself. It must have the paragraph heading "Data Availability Statement". All data supporting the results in the paper must be either: in the paper itself; uploaded as Supporting Information for Online Publication; or archived in an appropriate public repository. The statement needs to describe the availability or the absence of shared data. Authors must include in their Statement: a link to the repository they have used, or a statement that it is available as Supporting

Information; reference the data in the appropriate sections(s) of their manuscript; and cite the data they have shared in the References section. Whenever possible the scripts and other artefacts used to generate the analyses presented in the paper should also be publicly archived. If sharing data compromises ethical standards or legal requirements then authors are not expected to share it, but must note this in their Statement. For more information, see our Statistics Policy.

- Please include an Abstract Figure file, as well as the figure legend text within the main article file. The Abstract Figure is a piece of artwork designed to give readers an immediate understanding of the research and should summarise the main conclusions. If possible, the image should be easily 'readable' from left to right or top to bottom. It should show the physiological relevance of the manuscript so readers can assess the importance and content of its findings. Abstract Figures should not merely recapitulate other figures in the manuscript. Please try to keep the diagram as simple as possible and without superfluous information that may distract from the main conclusion(s). Abstract Figures must be provided by authors no later than the revised manuscript stage and should be uploaded as a separate file during online submission labelled as File Type 'Abstract Figure'. Please ensure that you include the figure legend in the main article file. All Abstract Figures should be created using BioRender. Authors should use The Journal's premium BioRender account to export high-resolution images. Details on how to use and access the premium account are included as part of this email.

EDITOR COMMENTS

Reviewing Editor:

While both reviewers recognised the clinical interest in the issue, Reviewer 1 requests additional discussion around the results and context, while Reviewer 2 requests significant methodological clarifications. Subject to suitable revision a resubmission could be reconsidered.

Senior Editor:

While both reviewers recognise the clinical interest in the issue, Reviewer 1 requests additional discussion around the results and context, while Reviewer 2 requests significant methodological clarifications.

REFEREE COMMENTS

Referee #1:

To the authors:

The manuscript provides a comprehensive overview of the current state of knowledge regarding antenatal hypoxia-ischemia (HI) and fetal heart rate (FHR) patterns as a biomarker for antenatal brain injury. The introduction provides a strong rationale for the study, highlighting the significant burden of perinatal mortality and morbidity associated with HI and the need for new diagnostic and prognostic biomarkers. The methods section is detailed and clear, allowing for replication of the study. The results section is well-organized and provides sufficient detail, including the use of figures to aid in understanding the findings.

However, it is my opinion there are a few areas where the manuscript could be improved:

The title could be more concise and clearer. Consider revising it to something like "Circadian patterns of FHRV in fetal sheep following antenatal HI: a potential biomarker for brain injury."

While the articles provided a great overview of the role of HIF in the circadian system during adult life, recent evidence supports that the circadian system and inflammation are interconnected and fully related to fetal development. Therefore, I think that more information could be included about the fetal circadian system and peripheral oscillators. It would be beneficial to include a deeper discussion on the circadian rhythm of the fetus, as well as the importance of fetal peripheral oscillators in regulating various biological processes during development, including hypoxia response and adrenal regulation as well. It is my opinion that maybe a discussion brief about the role of cytokines in this regard may help to the correct.

Additionally, the articles could benefit from a further explanation of how disruptions in the fetal circadian system can lead to developmental abnormalities and predispose the individual to various health conditions later in life. The inclusion of more research and evidence in this area would help strengthen the articles' overall argument and provide readers with a more comprehensive understanding of the topic.

However, I have identified a few areas where the manuscript could be improved. Firstly, I suggest revising the title to make it more concise and clearer, such as "Circadian patterns of FHRV in fetal sheep following antenatal HI: a potential biomarker for brain injury."

Furthermore, while the manuscript provides a great overview of the role of HIF in the circadian system during adult life,

recent evidence suggests that the circadian system and inflammation are interconnected and fully related to fetal development. Therefore, I believe that it would be beneficial to include a deeper discussion on the circadian rhythm of the fetus and the importance of fetal peripheral oscillators in regulating various biological processes during development, including hypoxia response and adrenal regulation. Additionally, a brief discussion about the role of cytokines in this regard may help to improve the manuscript's content.

Finally, I think that the manuscript could benefit from a further explanation of how disruptions in the fetal circadian system can lead to developmental malfunctions and predispose the individual to various health conditions later in life. The inclusion of more research and evidence in this area would help strengthen the manuscript's overall argument and provide readers with a more comprehensive understanding of the topic.

Overall, I believe that the manuscript has the potential to make a significant contribution to the field with some improvements.

Thank you for the opportunity to review your work, and I look forward to seeing your revised manuscript.

Referee #2 (please see attachment):

Lear et al. propose a study on the evaluation of FHRV and circadian rhythms after hypoxia-ischaemia.

The question is interesting. The team is recognized in experimental research and fetal physiology.

I have 2 main comments:

1/ Study population. The main question is about the study population. It comes from a first publication of the team (Lear et al., Brain communications, 2021). In this series where 9 preterm fetal sheep "sham" were compared to 9 with UCO, the authors found that delayed appearance of cystic injury was consistent with continuing tertiary evolution of necrotic cell death

In this study (table 3), 3 fetuses shown almost no lesions: one with "Focal cystic white matter lesions in the temporal lobe and first parasagittal gyrus of the parietal lobe; No ventriculomegaly or reduction in white matter area" and two with "No cystic lesions or ventriculomegaly and Moderate reduction in white matter area".

In this new study, only 7 lambs were included because of the inherent problems of this type of experimentation of difficulties in signal recording. The authors do not specify which lambs were included. It is essential to specify this in the "methods" section. Did the 7 fetal sheep have cerebral lesions? What was the individual evolution of HRV parameters?

2/ Methodology. The authors base their interpretation partly on the accelerations. Several important limitations are present:

- These are not clearly defined ("abrupt increase" "onset of acceleration to peak in < 30sec"). There is of course no consensual definition of fetal heart rate analysis in ewes, but this is a limitation to the reproducibility of the analysis of the tracings.

- Were the analyses done blindly? Since they were done a posteriori, based on visual analysis of accelerations, knowing the group is an important bias.

- Was the visual analysis done by one or more people?

The rest of the article is very well written. The discussion should be reduced.

END OF COMMENTS

Confidential Review

21-Feb-2023

Lear et al. propose a study on the evaluation of FHRV and circadian rhythms after hypoxia-ischaemia. The question is interesting. The team is recognized in experimental research and fetal physiology.

I have 2 main comments:

1/ Study population. The main question is about the study population. It comes from a first publication of the team (Lear et al., Brain communications, 2021). In this series where 9 preterm fetal sheep "sham" were compared to 9 with UCO, the authors found that delayed appearance of cystic injury was consistent with continuing tertiary evolution of necrotic cell death

In this study (table 3), 3 fetuses shown almost no lesions: one with "Focal cystic white matter lesions in the temporal lobe and first parasagittal gyrus of the parietal lobe; No ventriculomegaly or reduction in white matter area" and two with "No cystic lesions or ventriculomegaly and Moderate reduction in white matter area".

In this new study, only 7 lambs were included because of the inherent problems of this type of experimentation of difficulties in signal recording. The authors do not specify which lambs were included. It is essential to specify this in the "methods" section. Did the 7 fetal sheep have cerebral lesions? What was the individual evolution of HRV parameters?

2/ Methodology. The authors base their interpretation partly on the accelerations. Several important limitations are present:

- These are not clearly defined ("abrupt increase" "onset of acceleration to peak in < 30sec"). There is of course no consensual definition of fetal heart rate analysis in ewes, but this is a limitation to the reproducibility of the analysis of the tracings.
- Were the analyses done blindly? Since they were done a posteriori, based on visual analysis of accelerations, knowing the group is an important bias.
- Was the visual analysis done by one or more people?

The rest of the article is very well written. The discussion should be reduced.

13/05/2023

Contact details:

Professor Laura Bennet,
Head of Department
Chair of Perinatal Physiology
Department of Physiology,
Faculty of Medical and Health Sciences
The University of Auckland,
Private Bag 92019,
Auckland, New Zealand.
Tel (+64 9) 373 7599 ext. 84890
E-mail: l.bennet@auckland.ac.nz

To: The Editor,
Journal of Physiology

Re: Circadian patterns of heart rate variability in fetal sheep after hypoxia-ischaemia: a biomarker of evolving brain injury. Christopher A. Lear, Yoshiki Maeda, Victoria J. King, Simerdeep K. Dhillon, Michael J. Beacom, Mark I. Gunning, Benjamin A. Lear, Joanne O. Davidson, Peter R. Stone, Tomoaki Ikeda, Alistair J. Gunn, Laura Bennet.

Category: Original Article

Dear Editor,

Thank you for inviting a revision of this manuscript. We thank the referees for their time and effort reviewing our manuscript. We have addressed the editors and reviewers comments in full as detailed below. We trust that the manuscript is now ready for publication in your journal.

Laura Bennet and Alistair J. Gunn
On behalf of the authors

Referee #1:

The manuscript provides a comprehensive overview of the current state of knowledge regarding antenatal hypoxia-ischemia (HI) and fetal heart rate (FHR) patterns as a biomarker for antenatal brain injury. The introduction provides a strong rationale for the study, highlighting the significant burden of perinatal mortality and morbidity associated with HI and the need for new diagnostic and prognostic biomarkers. The methods section is detailed and clear, allowing for replication of the study. The results section is well-organized and provides sufficient detail, including the use of figures to aid in understanding the findings.

Reply: We appreciate this feedback, thank you.

However, it is my opinion there are a few areas where the manuscript could be improved:

The title could be more concise and clearer. Consider revising it to something like "Circadian patterns of FHRV in fetal sheep following antenatal HI: a potential biomarker for brain injury."

Reply: Thank you for this suggestion, however abbreviations are not allowed in the title and thus although the suggested alternative would be equally appropriate, it is no shorter once abbreviations are removed. We have tweaked the title slightly; hoping that this seems suitable.

While the articles provided a great overview of the role of HIF in the circadian system during adult life, recent evidence supports that the circadian system and inflammation are interconnected and fully related to fetal development. Therefore, I think that more information could be included about the fetal circadian system and peripheral oscillators. It would be beneficial to include a deeper discussion on the circadian rhythm of the fetus, as well as the importance of fetal peripheral oscillators in regulating various biological processes during development, including hypoxia response and adrenal regulation as well. It is my opinion that maybe a discussion brief about the role of cytokines in this regard may help to the correct.

Additionally, the articles could benefit from a further explanation of how disruptions in the fetal circadian system can lead to developmental abnormalities and predispose the individual to various health conditions later in life. The inclusion of more research and evidence in this area would help strengthen the articles' overall argument and provide readers with a more comprehensive understanding of the topic.

However, I have identified a few areas where the manuscript could be improved. Firstly, I suggest revising the title to make it more concise and clearer, such as "Circadian patterns of FHRV in fetal sheep following antenatal HI: a potential biomarker for brain injury."

Furthermore, while the manuscript provides a great overview of the role of HIF in the circadian system during adult life, recent evidence suggests that the circadian system and inflammation are interconnected and fully related to fetal development. Therefore, I believe that it would be beneficial to include a deeper discussion on the circadian rhythm of the fetus and the importance of fetal peripheral oscillators in regulating various biological processes during development, including hypoxia response and adrenal regulation. Additionally, a brief discussion about the role of cytokines in this regard may help to improve the manuscript's content.

Finally, I think that the manuscript could benefit from a further explanation of how

disruptions in the fetal circadian system can lead to developmental malfunctions and predispose the individual to various health conditions later in life. The inclusion of more research and evidence in this area would help strengthen the manuscript's overall argument and provide readers with a more comprehensive understanding of the topic.

Reply: Thank you for these thoughtful and helpful comments, we have made the following amendments to our manuscript to address these suggestions, while respecting reviewer 2's request to limit the length of our discussion:

"It is important to consider the possibility that altered or impaired circadian rhythms after perinatal HI, as identified in the present study may contribute to ongoing neurodevelopmental impairment and risk of systemic and metabolic disease. Even in the absence of neural injury, chronodysruption during gestation leads to long-lasting impairment of circadian gene regulation and function in the adrenal gland (Richter et al., 2018; Salazar et al., 2018; Mendez et al., 2019) and hippocampus (Vilches et al., 2014). Gestational chronodysruption is also associated with impaired glucose and adipose tissue regulation, likely increasing risk of metabolic disease in offspring (Halabi et al., 2021). Moreover, in rodents, gestational chronodysruption has been associated with increased circulating levels of the pro-inflammatory cytokines interleukin-1 β and interleukin-6 in later life (Mendez et al., 2019). This suggests the hypothesis that persisting chronodysruption may contribute to chronic neuroinflammation and impaired neurodevelopment, and conversely that re-establishing normal circadian patterns could improve neurodevelopmental and systemic outcomes (Mendez et al., 2022)."

Overall, I believe that the manuscript has the potential to make a significant contribution to the field with some improvements.

Thank you for the opportunity to review your work, and I look forward to seeing your revised manuscript.

Reply: Thank you for these supportive comments.

Referee #2 (please see attachment):

Lear et al. propose a study on the evaluation of FHRV and circadian rhythms after hypoxia-ischaemia.

The question is interesting. The team is recognized in experimental research and fetal physiology.

I have 2 main comments:

1/ Study population. The main question is about the study population. It comes from a first publication of the team (Lear et al., Brain communications, 2021). In this series where 9 preterm fetal sheep "sham" were compared to 9 with UCO, the authors found that delayed appearance of cystic injury was consistent with continuing tertiary evolution of necrotic cell death

In this study (table 3), 3 fetuses shown almost no lesions: one with "Focal cystic white matter lesions in the temporal lobe and first parasagittal gyrus of the parietal lobe; No ventriculomegaly or reduction in white matter area" and two with "No cystic lesions or ventriculomegaly and Moderate reduction in white matter area".

In this new study, only 7 lambs were included because of the inherent problems of this type of experimentation of difficulties in signal recording. The authors do not specify which lambs were included. It is essential to specify this in the "methods" section. Did the 7 fetal sheep have cerebral lesions? What was the individual evolution of HRV parameters?

Reply: Thank you for this important question. For clarity, in relation to Table 3 of doi: 10.1093/braincomms/fcab024, the two fetuses that were excluded in this study due to unusable ECGs were both from the 'Marked white matter atrophy and ventriculomegaly' subgroup. As such, the group in this study represented fetuses across the full spectrum of injury reported in this model. We have added this information to the methodology section.

In regards to the evolution of FHRV in individual fetuses in relation to their subgroup of macroscopic white matter injury, our study is unfortunately not powered to answer this question as it was not a primary outcome. For the reviewers interest we have undertaken exploratory analysis. Visual assessment of the timecourse FHRV data broadly suggests that the more severely injured fetuses showed more marked exacerbation of circadian rhythms, but with substantial overlap across subgroups. In order to quantify these patterns, we have correlated white matter area (as measured in our previous studies via histological and microscopic assessment) against the peak-to-peak amplitude of FHRV and physiological measures over the final week of recovery. Those data are provided below. On simple linear regression, none of these correlations were significant and there is substantial overlap between the groups.

Overall we believe these findings are too preliminary to include in the present manuscript and are unlikely to bring further clarity in addition to the data already presented. We are happy to be guided by the editor and reviewer if they wish for this data to be included in the manuscript.

○ Control ● HI

2/ Methodology. The authors base their interpretation partly on the accelerations. Several important limitations are present:

- These are not clearly defined ("abrupt increase" "onset of acceleration to peak in < 30sec"). There is of course no consensual definition of fetal heart rate analysis in ewes, but this is a limitation to the reproducibility of the analysis of the tracings.

- Were the analyses done blindly? Since they were done a posteriori, based on visual analysis of accelerations, knowing the group is an important bias.
- Was the visual analysis done by one or more people?

Reply: Thank you for this important methodological question. The accelerations were assessed visually by a single, obstetrically trained investigator (YM) who was blinded to the groups by coding of individual datafiles. We have added this additional information and clarified that we utilised human definitions of accelerations.

“Accelerations were defined as per the FIGO guidelines for human fetuses <32 weeks of gestational as abrupt (onset of acceleration to peak FHR in <30 seconds) increases in FHR above the baseline, of more than 10 bpm and lasting >10 seconds but less than 10 minutes.”

The rest of the article is very well written. The discussion should be reduced.

Reply: We sincerely appreciate this feedback, thank you.

References

- Halabi D, Richter HG, Mendez N, Kähne T, Spichiger C, Salazar E, Torres F, Vergara K, Seron-Ferre M & Torres-Farfan C. (2021). Maternal Chronodisruption Throughout Pregnancy Impairs Glucose Homeostasis and Adipose Tissue Physiology in the Male Rat Offspring. *Frontiers in endocrinology* **12**, 678468.
- Mendez N, Halabi D, Salazar-Petres ER, Vergara K, Corvalan F, Richter HG, Bastidas C, Bascur P, Ehrenfeld P, Seron-Ferre M & Torres-Farfan C. (2022). Maternal melatonin treatment rescues endocrine, inflammatory, and transcriptional deregulation in the adult rat female offspring from gestational chronodisruption. *Front Neurosci* **16**, 1039977.
- Mendez N, Torres-Farfan C, Salazar E, Bascur P, Bastidas C, Vergara K, Spichiger C, Halabi D, Vio CP & Richter HG. (2019). Fetal Programming of Renal Dysfunction and High Blood Pressure by Chronodisruption. *Frontiers in endocrinology* **10**, 362.
- Richter HG, Mendez N, Abarzua-Catalan L, Valenzuela GJ, Seron-Ferre M & Torres-Farfan C. (2018). Developmental Programming of Capuchin Monkey Adrenal Dysfunction by Gestational Chronodisruption. *BioMed research international* **2018**, 9183053.
- Salazar ER, Richter HG, Spichiger C, Mendez N, Halabi D, Vergara K, Alonso IP, Corvalán FA, Azpeleta C, Seron-Ferre M & Torres-Farfan C. (2018). Gestational chronodisruption leads to persistent changes in the rat fetal and adult adrenal clock and function. *J Physiol* **596**, 5839-5857.
- Vilches N, Spichiger C, Mendez N, Abarzua-Catalan L, Galdames HA, Hazlerigg DG, Richter HG & Torres-Farfan C. (2014). Gestational chronodisruption impairs hippocampal expression of NMDA receptor subunits Grin1b/Grin3a and spatial memory in the adult offspring. *PLOS ONE* **9**, e91313.

Dear Dr Gunn,

Re: JP-RP-2023-284560R1 "Circadian patterns of heart rate variability in fetal sheep after hypoxia-ischaemia: a biomarker of evolving brain injury" by Christopher Arthur Lear, Yoshiki Maeda, Victoria J. King, Simerdeep Kaur Dhillon, Michael J. Beacom, Mark I. Gunning, Benjamin A. Lear, Joanne Davidson, Peter Richard Stone, Tomoaki Ikeda, Alistair J. Gunn, and Laura Bennet

We are pleased to tell you that your paper has been accepted for publication in The Journal of Physiology.

Authors should note that it is too late at this point to offer corrections prior to proofing. The accepted version will be published online, ahead of the copy edited and typeset version being made available. Major corrections at proof stage, such as changes to figures, will be referred to the Editors for approval before they can be incorporated. Only minor changes, such as to style and consistency, should be made at proof stage. Changes that need to be made after proof stage will usually require a formal correction notice.

Yours sincerely,

Peter Kohl
Senior Editor
The Journal of Physiology

P.S. - You can help your research get the attention it deserves! Check out Wiley's free Promotion Guide for best-practice recommendations for promoting your work at www.wileyauthors.com/eeo/guide. You can learn more about Wiley Editing Services which offers professional video, design, and writing services to create shareable video abstracts, infographics, conference posters, lay summaries, and research news stories for your research at www.wileyauthors.com/eeo/promotion.

IMPORTANT NOTICE ABOUT OPEN ACCESS: To assist authors whose funding agencies mandate public access to published research findings sooner than 12 months after publication, The Journal of Physiology allows authors to pay an Open Access (OA) fee to have their papers made freely available immediately on publication.

You can check if your funder or institution has a Wiley Open Access Account here: <https://authorservices.wiley.com/author-resources/Journal-Authors/licensing-and-open-access/open-access/author-compliance-tool.html>.

REFeree COMMENTS

Referee #1:

Thank you for considering our feedback on your manuscript. I appreciate your willingness to incorporate the suggested improvements, as they will enhance the clarity and impact of your study. Based on my assessment, I find the manuscript to be strong in several key aspects:

Impact on the area of research: Your study has the potential to make a significant impact in the field, providing valuable insights into the role of the circadian system in fetal development. The findings have implications for fetal and neonatal physiological mechanisms, contributing to our understanding of how a broad knowledge in fetal circadian rhythm may help to prevent long-term effects in the offspring.

Insight into physiological mechanisms: Your research offers important insights into the physiological mechanisms underlying.

Therefore, in summary, the current manuscript provides:

The originality of the research: The novelty and originality of your research are commendable. Your study takes a unique approach to address the research question, offering fresh perspectives, and pushing the boundaries of knowledge in the field.

Study design and robustness of the experimental data: Your study design is thorough and well-executed, allowing for reliable data collection. The experimental data presented is robust and supported by detailed descriptions, contributing to the credibility of your study.

Validity of the conclusions: The conclusions drawn from your study are valid and well-supported by the data and analysis presented. They align with the objectives outlined in the introduction and contribute to the broader understanding.

In conclusion, I believe your manuscript has a strong impact on the area of research, provides valuable insights into physiological mechanisms, exhibits originality, demonstrates a robust study design with reliable experimental data, and draws valid conclusions. These strengths position your study for publication and contribute to the advancement of knowledge in the field of fetal physiology and the role of the circadian system in this regard.

Referee #2:

Thank you for all the changes and the response to our reviewing.